# Advances in Membranous Nephropathy

**DOI:** 10.3390/jcm10040607

**Published:** 2021-02-05

**Authors:** Pierre Ronco, Emmanuelle Plaisier, Hanna Debiec

**Affiliations:** 1Unité Mixte de Recherche S1155, Institut National de la Santé et de la Recherche Médicale, Sorbonne Université, Université Pierre et Marie Curie Paris 06, Hôpital Tenon, 75020 Paris, France; emmanuelle.plaisier@aphp.fr (E.P.); hanna.debiec@upmc.fr (H.D.); 2Reference Center of Rare Disease-Idiopathic Nephrotic Syndrome, Hôpital Tenon, 75020 Paris, France; 3Department of Nephrology, Centre Hospitalier du Mans, 72000 Le Mans, France; 4Day Hospital of Nephrology, Hôpital Tenon, Assistance Publique-Hôpitaux de Paris, 75020 Paris, France

**Keywords:** membranous nephropathy, randomized controlled trials, rituximab, KDIGO recommendations, PLA2R, THSD7A, exostosins 1/2, NELL-1, NCAM-1, Semaphorin 3B

## Abstract

Membranous nephropathy (MN) is a rare auto-immune disease where the glomerulus is targeted by circulating auto-antibodies mostly against podocyte antigens, which results in the formation of electron-dense immune complexes, activation of complement and massive proteinuria. MN is the most common cause of nephrotic syndrome in adults leading to severe thrombotic complications and kidney failure. This review is focused on the recent therapeutic and pathophysiological advances that occurred in the last two years. For a long time, we were lacking a head-to-head comparison between cyclophosphamide considered as the gold standard therapy and other medications, notably rituximab. Substantial progress has been achieved owing to three randomized controlled trials. MENTOR (Membranous Nephropathy Trial of Rituximab) and STARMEN (Sequential Therapy with Tacrolimus and Rituximab in Primary Membranous Nephropathy) conclusively established that calcineurin inhibitor-based regimens are slower to result in an immunologic response than rituximab or cyclophosphamide, achieve fewer complete clinical remissions, and are less likely to maintainremission. Rituximab Versus Steroids and Cyclophosphamide in the Treatment of Idiopathic Membranous Nephropathy (RI-CYCLO) suggested that competition between cyclophosphamide and rituximab remains open. Given the technological leap combining laser microdissection of glomeruli and mass spectrometry of solubilized digested proteins, four “new antigens” were discovered including NELL-1 and Semaphorin 3B in so-called primary MN, and exostosins 1 and 2 and NCAM 1 in lupus MN. NELL-1 is associated with about 8% of primary MN and is characterized by segmental immune deposits and frequent association with cancer (30%). Semaphorin 3B-associated MN usually occurs in children, often below the age of two years, where it is the main antigen, representing about 16% of non-lupus MN in childhood. Exostosins 1/2 and NCAM 1 are associated with 30% and 6% of lupus MN, respectively. Exostosins 1/2 (EXT1/2) staining is associated with a low rate of end-stage kidney disease (ESKD) even in mixed classes III/IV+V. These findings already lead to revisiting the diagnostic and therapeutic algorithms toward more personalized medicine.

Membranous nephropathy (MN) is a rare disease affecting the glomerular capillary wall, which plays a key role in controlling permeability to proteins. It accounts for about 20% of cases of nephrotic syndrome in adults and is the leading glomerulopathy recurring after kidney transplantation. Histologically, MN is characterized by the accumulation of electron-dense subepithelial immune deposits, which cause a membrane-like thickening with the formation of spikes. The immune deposits consist of immunoglobulin (Ig) G, the relevant antigens, and the membrane attack complex of complement. There is no infiltration of the glomerulus by inflammatory cells, most likely because the immunological conflict takes place outside the glomerular basement membrane. Despite a common histopathological pattern, MN is a heterogeneous disease, occurring either in the absence of an associated disease (80% of cases) or in association with clinical conditions such as infections (hepatitis B), lupus erythematosus, cancer, or drug intoxication, thus defining the so-called primary and secondaryMN, respectively. Heterogeneity is also illustrated by the variable outcome with 40% of patients undergoing spontaneous remission while about 20% will evolve toward severe kidney failure requiring renal replacement therapy [1].

The treatment armentarium for primary forms of MN has long been dominated by alkylating agents and calcineurin inhibitors (CNIs). The demonstration of the autoimmune nature of the disease with the identification of PLA2R has stimulated the development of more targeted and less toxic anti-CD20-based therapies, leading to three randomized clinical trials (RCTs) with CNI or cyclophosphamide as a comparator in the last two years. Because of the focus on anti-PLA2R antibody titers for the monitoring of treatment, considerable efforts have been made to identify additional antigens in the roughly 20% PLA2R-negative MN. The technological leap combining laser microdissection of glomeruli and mass spectrometry of solubilized digested proteins has allowed theidentification offour“new” antigens, and others are in the pipeline.The first part of the review describes the new therapeutic advances with rituximab-based protocols and PLA2R-based monitoring, which were initiated by the GEMRITUX trial. These data have been incorporated in the KDIGO 2021 Guideline for the Management of Glomerular Diseases that will be released in the first semester of 2021. The second part presents new pathophysiological knowledge beyond PLA2R with the discovery of the “new” antigens that question the classification of the disease between primary and secondary forms and already have an impact on diagnostic algorithms.

## 1. New Advances in Treatment

Although the combination of corticosteroids and an alkylating agent is the only therapy so far shown to preserve kidney function in the long-term [2,3], anti-CD20 biotherapy, particularly rituximab [4], has become increasingly popular as first line therapy because of the toxicity of the alkylating agents and the long-term renal toxicity of calcineurin inhibitors (CNIs) and the high risk of relapsewhen they are used as monotherapy with or without prednisone.

The GEMRITUX trial [5] was a multicentric randomized controlled trial (RCT) performed in France, in which 77 patients with normal renal function were randomized to either rituximab (375 mg/m^2^ on days one and eight) or continuation of non-immunosuppressive anti-proteinuric treatment after a 6-month run-in period of anti-proteinuric therapy in each arm. In the follow-up period after the first 6 months, in which the difference was not significant, the remission rate in terms of proteinuria was of 65% in the rituximab group, compared to 34% in the control group. Anti-PLA2R antibody titers were significantly reduced as early as three months in the rituximab group. This confirmed that a reduction in antibody titers precedes the reduction in proteinuria and suggested that monitoring of titers could be used to predict the need for a repeat infusion. A retrospective cohort study comparing rituximab to glucocorticoids and oral cyclophosphamide according to a well-defined standardized protocol showed no difference at 5 years in the rate of complete clinical remission, but a better safety profile in the rituximab-treated group [6]. However, until recently there was no head-to-head comparison between alkylating agents, CNIs and rituximab in RCTs.

New trials since 2019: MENTOR, STARMEN, RI-CYCLO (Table 1).

The MENTOR trial [7] was the first RCT conducted in North America comparing rituximab to cyclosporine. The trial included 130 patients with at least 5 g/day of proteinuria and creatinine clearance >40 mL/min/1.73 m^2^. Patients were treated with rituximab (two infusions, 1000 mg each, 14 days apart, repeated at six months in case of partial response) or cyclosporine (at a dose of 3.5 mg per kilogram of body weight per day for 12 months then progressively tapered). At 12 months, 39 of 65 patients (60%) in the rituximab group and 34 of 65 (52%) in the cyclosporine group achieved complete or partial remission. However, at 24 months, 39 patients (60%) in the rituximab group remained in remission, while the rate of remission in the cyclosporine group had dropped to 20% (13/65) due to a large number of relapses after the cyclosporine was tapered and stopped. Furthermore, 35% of the rituximab group were in complete remission, compared to none who had received cyclosporine. Therefore, while rituximab was found to be non-inferior to cyclosporine in inducing remission at 12 months, it was statistically superior at 24 months in terms of maintaining long-term remission. In addition, among the 74% of patients with anti-PLA2R-associated MN, the decline in anti-PLA2R autoantibody titers was more rapid and intense in the rituximab group than in the cyclosporine group. At the end of the study more cyclosporine-treated patients suffered ≥50% decline in creatinine clearance compared to the rituximab group.Both agents were comparably safe and well tolerated.

The STARMEN trial [8] is a European RCT that compared the efficacy of a sequential therapy with tacrolimus and rituximab to that of a cyclical alternating treatment with a modified Ponticelli protocol in inducing persistent clinical remission. This open-label controlled trial enrolled 86 patients with primary MN and persistent nephrotic syndrome after a run-in period of six-months.Patients had at least 4 g/day of proteinuria and anestimated glomerular filtration rate(eGFR) > 45 mL/min/1.73 m^2^. Forty-three patients in each group were assigned to receive six-month cyclical treatments with corticosteroid (total cumulative dose of oral methylprednisolone, 3.4+/−0.9g and of intravenous methylprednisolone, 8.2+/−1.4 g) and cyclophosphamide (total cumulative dose, 10+/−3.5 g) or sequential treatment with tacrolimus (full-dose for six months and tapering for another three months) and rituximab (one g at month six). Seventy-seven percent of the patients were PLA2R positive.

The primary outcome was complete or partial remission of nephrotic syndrome at 24 months. This composite outcome occurred in 36 patients (84%) in the corticosteroid-cyclophosphamide group and in 25 patients (58%) in the tacrolimus-rituximab group. Complete remission at 24 months occurred in 26 patients (60%) in the corticosteroid-cyclophosphamide group and in 11 patients (26%) in the tacrolimus-rituximab group. Anti-PLA2R titers showed a significant decrease in both groups but the proportion of anti-PLA2R-positive patients who achieved immunological response (depletion of anti-PLA2R antibodies) was significantly higher at three and six months in the corticosteroid-cyclophosphamide group (77% and 92%, respectively), as compared to the tacrolimus-rituximab group before the infusion of rituximab (45% and 70%, respectively). Relapses occurred in one patient in the corticosteroid-cyclophosphamide group, and three patients in the tacrolimus-rituximab group. Serious adverse events were similar in both groups. Thus, treatment with corticosteroid-cyclophosphamide induced remission in a significantly greater number of patients with primary MN than tacrolimus-rituximab. However, data should be interpreted with caution because of several potential biases such as the relatively low percentage of males among patients treated with the adapted Ponticelli protocol (55% vs. 72% in the tacrolimus-rituximab group), and trends toward higher PLA2R-Ab titers, a higher interquartile range of proteinuria and lower interquartile range of serum albumin in the tacrolimus-rituximab group. The superiority of corticosteroid-cyclophosphamide over tacrolimus-rituximab might also be related to the relatively small dose of rituximab given to patients (only 1 g, instead of 2 g or even 4 g in total in MENTOR).

In summary, MENTOR and STARMEN show that CNI-based regimens are slower to result in an immunologic response than rituximab or cyclophosphamide and achieve fewer complete clinical remissions. CNI-based treatments are less likely to maintain a patient in remission unless something like rituximab is added, but this does not seem to improve efficacy beyond rituximab alone. However, it is important to note that the addition of rituximab was delayed by six months in STARMEN.

RI-CYCLO [9] is a pilot randomized controlled trial that was aimed to assess the recruitment potential using a multi-site design (11 centers mostly in Italy) and to obtain estimates of the effects of rituximab (1 g, 15 days apart) compared to that of a cyclical alternating treatment with a modified Ponticelli protocol. The trial enrolled 74 patients with primary MN and persistent nephrotic syndrome after a run-in period of three months.Patients had at least 3.5 g/day of proteinuria and an eGFR ≥30 mL/min/1.73 m^2^. The primary outcome was complete remission of proteinuria at 12 months. Other outcomes included complete or partial remission at 24 months and adverse events.

At 12 months, only 6/37 (16%) patients in the rituximab arm achieved a complete remission as compared to 12/37 (32%) patients in the cyclical regimen arm, while 23/37 (62%) patients in the rituximab arm and 27/37 (73%) in the cyclical regimen arm had a complete or partial remission. At 24 months, probabilities of complete and partial remission were comparable between the 2 groups. Serious adverse events occurred in 19% of patients in the rituximab arm and in 14% in the cyclical regimen arm. The authors concluded that this pilot trial failed to detect any signal of more benefit or less harm of rituximab vs. cyclical Ponticelli regimen in the treatment of primary MN. However, the cyclical regimen tended to induce complete remission earlier, and while non-significantly different, the point estimate of one-year probability of complete remission was lower in the rituximab arm. Of note, results of this trial should be interpreted with caution because of possible bias linked to the very long duration of recruitment (84 months) in 11 centers.

## 2. KDIGO Recommendations

The KDIGO 2021 Guideline for the Management of Glomerular Diseases are in press [10]. Considerable changes have occurred since the previous issue of the KDIGO in 2012 [11].

While a “wait and see”period is still recommended in nephrotic patients treated with anti-proteinuric agents at maximally tolerated doses owing to the high rate of spontaneous remission [12], it is now well accepted that the period of six months can be shortened in patients with a high risk of complication or kidney disease progression (Figure 1).There is currently no model that establishes a risk score, but in clinical practice, risk should be considered as a combination of factors (e.g., high proteinuria in patients with low antibody titers is judged differently than high proteinuria in the presence of high antibody titers).

The algorithm presented in Figure 1 stratifies the indications and modalities of therapy according to the degree of risk. In high-risk patients, choice of treatment depends on a number of variables including patient characteristics, drug availability, patient and physician preferences, cost and reimbursement policies, and the specific side effect profile of each drug. Low-risk patients should not be treated with immunosuppressive agents. In moderate- to high-risk patients, there are two important changes compared to KDIGO recommendations in 2012 [11]. First, rituximab is now on the frontline with cyclophosphamide in patients at risk although the long-term benefit on kidney function of anti-CD20 therapy remains to be established and the efficacy to induce immunological remission seems less in patients with very high titers of antibody (third tertile) [13]. The second change concerns CNIs. The MENTOR trial [7] has clearly confirmed the high rate of relapse in patients treated with cyclosporine. Given the observation that sustained remission are associated with improved kidney survival [14] and their lesser immunosuppressive effect on PLA2R-Ab, CNIs would not seem to be the best therapy for primary MN. Their use in moderate-risk patients who will most likely develop spontaneous remission may be justified to shorten the period of proteinuria. In higher risk patients, it is recommended that they should not be used as monotherapy but in combination with rituximab or another anti-CD20 drug. In the STARMEN trial [8], rituximab was given after six months of tacrolimus which seemed to reduce the rate of relapse. In very high-risk patients with rapid deterioration of renal function, there is insufficient evidence that rituximab used in standard doses prevents development of ESRD, and cyclophosphamide is usually preferred. However, rituximab seems effective and reasonably safe in PLA2R-associated MN in patients with stage-4 or -5 chronic kidney disease. In those patients, immunological remission was associated with a good clinical outcome [15].

## 3. Perspectives

### 3.1. Decreasing Toxicity of Cyclophosphamide and Alternate Regimens

A major dilemma in patients with MN is efficacy versus toxicity. In theSTARMEN trial [8], there were no differences in the number of serious adverse events among patients treated with corticosteroid-cyclophosphamide as compared to those who received tacrolimus followed by rituximab. However, cumulative doses of cyclophosphamide were lower in STARMEN than in previous studies [6,16]. As recommended in the KDIGO 2021 Guideline for the Management of Glomerular Diseases [10], the cumulative dose should not exceed 10g if preservation of fertility is required, and 25 g to limit risk of malignancies. There are several ways to decrease toxicity of cyclophosphamide-based regimens. The first is to employ lower doses of cyclophosphamide as suggested by observational studies using non-cyclical schemes of cyclophosphamide-based treatments MN [17]. The second possibility is the use of intravenous pulses as a substitute for oral cyclophosphamide in patients treated with the alternating cyclical regimen of corticosteroids and cyclophosphamide [18,19]. A third, controversial possibility is the combination of lower doses of cyclophosphamide with rituximab as advocated by some in the treatment of vasculitis [20].

On the other hand, as in many protocols now used in auto-immune diseases, lower doses of corticosteroids should be considered to reduce side effects without decreasing the effectiveness of alternating treatments. Future trials will need to address these important questions.

Whatever the adaptation to the initial Ponticelli protocol, it should be tailored to the level of PLA2R antibody (PLA2R-Ab) since it is unnecessary and dangerous to overexpose patients who have achieved immunological remission to potentially toxic drugs.

### 3.2. Improving the Use of Anti-CD20 Antibodies

A major pitfall of treatments using antibody-based biotherapies is the loss of antibody in the urine in severely nephrotic patients. A recent study comparingtwo protocols of rituximab, i.e., the one used in the GEMRITUX trial (2 × 375 mg/m^2^, one week interval) and the infusion of 2 × 1 g at one-week intervals, showed a better outcome with the latter although this has to be confirmed in future studies [21]. A retrospective study showed that all patients that failed to achieve immunological remission after a first course of rituximab achieved clinical remission after a second course [22]. The KDIGO 2021 Guideline for the Management of Glomerular Diseases [10] recommend to re-infuse rituximab at six months after the first infusion in patients who have not achieved immunological remission. To reinforce efficacy of rituximab, some investigators recommend starting with CNIs owing to their hemodynamic effects including a decrease in GFR to reduce the loss of rituximab in the urine. This approach was chosen by the STARMEN investigators who infused rituximab six months after starting tacrolimus [8]; however, the time between the two treatments exceeded the time required for a hemodynamic effect, and tacrolimus also had an immunosuppressive effect as judged by the decrease of anti-PLA2R level, albeit weaker than cyclophosphamide.

New anti-CD20 humanized antibodies are now on the market. They have several advantages over rituximab such as low risk of immunization against the monoclonal antibody and prolonged B-cell depletion.Several case reports and case series have shown that these new antibodies (ofatumumab, obinutuzumab, ocrelizumab) could be effective in refractory or multi-relapsing MN as well as in the patients treated with rituximab that developed serum sickness [23,24,25,26]. Since in about 30% to 40% of patients, rituximab fails to induce remission of MN and relapses are frequent, these more potent biotherapies will most likely take a substantial part in severe forms of MN in a near future.

### 3.3. New Therapeutic Approaches: Anti-Baff Therapy (Belimumab), Anti-Plasma Cell Therapy, Immunoadsorption, Anti-Complement Therapy

These new treatments are justified by the relatively low rate of success around 60% at 24 months of rituximab-based therapies compared to 84% with cyclophosphamide-based therapy but at the cost of more Severe Adverse Events (SAEs) (see Table 1, MENTOR and STARMEN).

Belimumab is a human IgG1-lambda monoclonal antibody that inhibits the B-cell activating factor (BAFF). It was approved for treatment of SLE and in a recent two-year RCT, it was shown that the patients who received belimumab plus standard therapy had a better renal response than those who received standard therapy alone [27]. Belimumab was used with anti-proteinuric agents in an open-label, prospective, single-arm studying 14 patients, of whom 11 completed the study [28]. Treatment induced a reduction of proteinuria and levels of circulating PLA2R-Ab by 86% and 97%, respectively. Nine participants achieved partial (*n* = 8) or complete (*n* = 1) remission. However, the decrease of PLA2R-Ab was more rapid in the rituximab-treated patients in GEMRITUX trial than in the belimumab study. Because rituximab increases the circulating levels of BAFF, some have advocated combination therapy with belimumab [29].

Although rituximab represents a major breakthrough in the treatment of many autoantibody-mediated diseases, its efficacy may be limited by the involvement of CD19-/CD20−/CD38+/CD138+ long-lived memory plasma cells that are niched naturally in the bone marrow and ectopically in the native or inflamed kidney. These nonproliferating plasma cells lacking CD19 and CD20 markers provide the basis for humoral memory and refractory autoimmune diseases. They may explain the limited rate of sustained complete remission in patients treated with rituximab [30]. Because these cells are targeted by anti-CD38 antibodies, the results of ongoing trials using anti-CD38 antibodies are eagerly awaited.

Immunoadsorption and plasmapheresis have been used by several groups to accelerate the depletion of circulating THSD7A- and PLA2R-Ab in patients with severe MN [31,32]. Only small series or cases have been reported. For the time being, we think the only indication is refractory disease and this should be discussed with a reference center.

Apart from immunosuppressive strategy, drugs aimed at reducing complement activation will be a part of the therapeutic armentarium in a near future. It is generally considered that both the lectin and alternative pathways are activated resulting in the formation of the terminal membrane attack complex of complement, which at least in experimental models, is the major mediator of proteinuria. There are two windows of opportunities for anti-complement therapy, early before immunosuppressive drugs reach full efficacy, and later in the patients with partial or no remission. Several drugs targeting different components of the lectin and alternative pathways are in early phase trials.

## 4. New Pathophysiological Advances: The Third Antigenic Revolution

Three waves of discovery punctuated the success story of antigen identification in MN.

The first one in 2002 led to the characterization of neutral endopeptidase (NEP) antigen in a rare subset of infants born with MN [33]. The disease developed because the mother was deficient in NEP due to a truncating mutation in the *MME* gene coding for NEP [34], responsible for allo-immunization during pregnancy and transplacental transfer of antibodies to the fetus in the last trimester of pregnancy. This finding paved the way for the second wave of discoveries that identified the major antigen in adult MN, PLA2R [35] followed by THSD7A [36], involved in 70–80% and less than 5% of primary MN respectively. Although these antigens were initially thought to be specific for primary MN, it was further shown that PLA2R-related MN could be associated with replicating hepatitis-B virus infection [37] and clinically active sarcoidosis [38]. In a few case reports, THSD7A-related MN seems to be clearly linked to cancer because the antigen was found in tumor cells and the patients achieved remission after efficacious anticancer therapy [39].

### 4.1. 2019: A New Turn in the Discovery of MN Antigens: Laser Microdissection of Glomeruli and Mass Spectrometry

A major technological leap based on laser microdissection of glomeruli followed by mass spectrometry (MS) identification of trypsin digested proteins was achieved by Sethi et al. at the Mayo Clinic [40]. A substantial difference with the immunochemical methods previously used for PLA2R and THSD7A discovery is that these authors started from paraffin-embedded biopsies, instead of sera, which allows identification of the antigen even in patients with immunologically inactive diseases. This technological leap resulted in the characterization of four additional antigens/biomarkers in 2019 and 2020, exostosins 1/2, NELL-1, semaphorin 3B and NCAM-1 [40,41,42,43], which led Hayashi and Beck to write that “technological advances have allowed for the demonstration of Moore’s law (a doubling every two years in the number of transistors that can be fit onto a computer chip) in the field of MN, and that even more antigens can be expected in the near future” [44].

Starting from PLA2R negative biopsies, which represent about 20% of all primary MN, the workflow of experiments first included a pilot cohort of a few number of biopsies where the MS spectra of trypsin digested micro-dissected glomeruli were compared between PLA2R negative and PLA2R positive biopsies, and identification of the putative antigen was followed by antigen detection in subepithelial immune deposits by immunohistochemistry (IHC). The second step was the screening by IHC of a large discovery cohort including a high number of controls, followed by MS confirmation of the nature of the antigen in the available biopsies. While the cohorts screened in the first two steps were from the Mayo Clinic, the third step enrolled European cohorts for replication of the data. Exostosins 1/2 were the first “antigens” identified using this approach. The molecular structures of these “new” antigens are shown in Figure 2 where they are compared to those of PLA2R and THSD7A.

### 4.2. Exostosins 1/2 (EXT1/2) and NCAM-1 Are Associated with Auto-Immune Diseases

EXT1/2 protein was identified in five of 15 PLA2R negative biopsies of the pilot cohort and in 21 of 209 PLA2R negative biopsies (201 non-lupus and eight lupus glomerulonephritis) of the discovery cohort, while 102 controls, including 13 proliferative lupus nephropathy, 47 PLA2R positive MN and 42 other types of glomerulonephritis, were all negative.In a replication cohort of 48 patients, three out of 16 patients with “primary” MN were EXT1/2 positive but they all had signs of autoimmunity on chart review and two patients later developed full-blown clinical lupus; 8 of 18 patients with class V lupus MN but only one of 14 patients with a mixed class stained positive for EXT1/2 [40].

Unlike the patients with primary MN, patients with EXT1/2 positive MN are young with a mean age of 35 years, 81% are females, 71% have signs of autoimmunity such as positive anti-nuclear, anti–double-stranded DNA, anti-SSA/SSB, or anti-ribonucleoprotein antibodies, and a clinical diagnosis of lupus could be posed in 35% of them. Biopsy findings revealed features suggestive of a secondary MN related to autoimmune disease in most patients. These atypical findings for a primary MN included frequent staining for C1q, IgA and/or IgM in the absence of a true full-house pattern, subendothelial and mesangial deposits, and tubuloreticular inclusions in endothelial cells on electron microscopy. Furthermore, IgG1 was the dominant IgG subclass with spectral counts significantly greater than for IgG4, the prevailing subclass in PLA2R-associated MN. Taken together, these findings suggest that EXT1/2 represent potential biomarkers or target antigens in secondary autoimmune MN [40].

EXT1 and EXT2 form a heterodimeric enzyme called glycosyl transferase which adds glycosyl residues to the protein backbone of proteoglycans [45]. This explains why the two proteins are found together both by MS and IHC. However, the term of antigen cannot be used as yet because antibodies were not detected in patients’ sera incubated with the recombinant proteins used under native conditions. Among other hypotheses, reactivity might be directed to an epitope which is only present in the enzyme produced by glomerular cells, while we used human eukaryoticrecombinant proteins produced in vitro in our blot assays.

Whether EXT1/2 are true antigens or not, recent studies from Sethi’s team [46] and from my team suggest that these biomarkers are detected in about 30% of pure lupus MN (class V) and are usually absent in other classes of lupus glomerulonephritis without subepithelial deposits. EXT1/EXT2-positive disease appears to represent a subgroup with lower chronicity indices and lower rates of progression to ESKD [46]. Because of the specificity of these biomarkers for lupus, EXT1/EXT2 staining prompts to anticipate later development of lupus disease in young female patients with a diagnosis of “primary” MN.

NCAM-1 was identified by Larsen’s group [43] using the same approach as Sethi et al. [40], but in addition they performed protein G immunoprecipitation studies from frozen biopsies. NCAM-1 was found in 20 biopsies of MN, including 5 with a proliferative component, and it was shown to colocalize with IgG within glomerular immune deposits. Unlike EXT1/2, antibodies to recombinant NCAM-1 were detected in the patients’ sera. NCAM1 was detected in 6.6% (14/212) of membranous lupus nephritis with or without proliferative changes and 2.0% (2/101) of primary MN. In the same study, 15.8% (33/209) of lupus MN stained positive for EXT2, a lower rate than that observed by Sethi et al. [40], probably because of the inclusion of mixed classes with a proliferative component.

Clinical and biopsy findings were similar to those of patients with EXT1/2-associated MN, with an average age of 34 years, 70% of females, frequent staining for IgA, IgM and C1q, variable staining of IgG subclasses but without predominance of IgG4 in most studied cases, and mesangial deposits by electron microscopy. Neuropsychiatric disease occurred in 8/20 (40%) patients possibly related to NCAM-1 expression in the central nervous system [47].

NCAM1 is a member of the Ig superfamily of proteins (MW (Molecular Weight), 120 kDa). In adults, NCAM1 expression is seen at high levels within the central nervous system, peripheral nerves, thyroid and adrenal gland, heart, stomach, and cells within the immune system including natural killer cells, γΔ T cells, activated CD8+ T cells, and dendritic cells. Interestingly, it could not be detected in podocytes in “normal” kidney biopsiesor biopsies from non-NCAM-1 MN while it was found in urinary exosomes [48] and its level in the urine was correlated with disease activity in a subset of lupus patients with proliferative lupus nephritis [49]. This raises the possibility that a soluble form of NCAM-1 might be involved in the pathogenesis of the disease.

In summary, EXT1/2 and NCAM-1 are the first biomarkers of lupus MN. They both are associated with subepithelial immune deposits, suggesting mechanisms of injury different in lupus MN from other forms of nephropathy. Further studies are needed to identify the antigens involved in the two-thirds of cases negative for both EXT1/2 and NCAM and to unravel the pathogenesis.

### 4.3. Neural Epidermal Growth Factor-Like 1 Protein (NELL-1) Is Associated with “Primary” MN and Cancer-Related MN

NELL-1 was identified in six of 35 PLA2R negative biopsies of the pilot cohort and in 23 of 91 PLA2R negative biopsies of the discovery cohort, while MS failed to detect NELL-1 in 23 PLA2R-positive MN and 88 controls [42]. Thus, 29 of 126 (23%) PLA2R negative biopsies were positive for NELL-1. Five NELL-1 positive cases out of 84 PLA2R and THSD7A negative biopsies were further identified in two replication cohorts from France and Belgium, making a total of 34 positive cases out of 210 (16.2%) biopsies that stained negative for PLA2R. By IHC, bright, granular staining of subepithelial immune deposits were seen in all NELL-1 positive cases although few cases showed segmental staining and many cases showed incomplete staining with negative capillary loops confirmed by electron microscopy. IgG and NELL-1 were strictly colocalized in the deposits by confocal fluorescence microscopy. By Western blot with human recombinant NELL1, anti-NELL-1 antibodies were detected in the blood under non-reducing conditions, while no reactivity was seen in control sera and under reducing conditions. Antibodies recognized the dimeric and trimeric forms of NELL-1 that resolved in a non-reactive monomer after reduction.

Clinical and biopsy findings showed features of primary MN with a mean age of 63 years and a slight male preponderance but differed by the IgG subclass pattern and the association with cancer in the validation cohorts. All four IgG subclasses were detected by MS, with IgG1 being the most abundant and IgG4 the least. While no case of cancer was detected in the Mayo cohort of 126 patients, four of the five patients with NELL-1 associated MN in the replication cohorts developed a cancer discovered at the time, or a few months after, the diagnosis of MN. This finding and the apparent rarity of the disease in Europe led us to consider that NELL-1 associated nephropathy might have different geographical prevalence and etiologies. Types of associated cancer included epidermoid lung cancer, metastatic pancreatic carcinoma, metastatic breast cancer, and infiltrating urothelial carcinoma. In one patient, chemotherapy induced remission of the cancer was associated with complete remission of the nephrotic syndrome and disappearance of antibodies.

These observations were largely confirmed by Caza et al. [50] who confirmed that NELL-1 could be the first antigen in malignancy associated MN. In their series of 91 patients, they found that IgG deposits were global but incomplete or segmental in 93.4% of cases, IgG1 subclass was present in all cases while IgG4 was detected in only 54%, and a cancer was diagnosed in 33% of cases. When available, the tumor biopsy stained positive for NELL-1. A majority of NELL1-associated MN patients with a history of malignancy had concurrent proteinuria and ongoing malignancy.

Caza et al. [50] also analyzed the distribution of the various antigens in a cohort of 1378 biopsies of “primary” MN collected over a five-year period at Arkana laboratories.Of the 111 biopsies of cancer-associated MN, 42 were with unknown antigen, 35 were PLA2R positive, 30 were NELL1 positive, and four were THSD7A positive. When the prevalence of cancer was analyzed per subcategory of MN identified by antigen, NELL-1 came first (33%) followed by THSD7A (11%) while PLA2R-positive cases accounted for only 4%. Thus, the finding of NELL-1 in a biopsy should trigger a detailed workup in search of a malignancy as recently discussed for PLA2R negative patients [51]. Although NELL-1 is not rarely expressed in tumors, only a few patients will develop MN, therefore further studies are needed to determine the genetic background and the triggering event.

Pure segmental MN is a rare entity that was recently reviewed by Kudose et al. [52]. Only 50 cases representing 2.5% of all MN biopsies were identified at Columbia University from January 2010 to October 2018. Staining for NELL-1 was positive in five of 17 cases (29%) available for study while staining for PLA2R, THSD7A, and EXT1 was negative. Thus NELL-1 appears to be the first antigen in segmental MN.

NELL-1 is a secreted, 90-kDa protein expressed in osteoblasts and promoting bone regeneration [53]. While it is overexpressed in patients with craniosynostosis, its expression in the kidney is weak and it is virtually absent from podocytes except in pathological conditions such as PLA2R-related MN. This finding suggests that NELL-1 may be present at very low levels in podocytes and induced by external stimuli or epigenetic modifications as is the case for PLAR1 in cancer cells [54]. Important questions remain about ultrastructural localization, the role of anti-NELL-1 antibodies, and mechanisms of immunization.

### 4.4. Semaphorin 3B (Sema3B) Is Associated with Early Childhood MN

For this study, additional cohorts of children recruited in Rome and Paris were screened because initial findings in the cohorts screened at Mayo Clinics and in Paris mostly composed of adult patients and indicated that the disease could be more frequent in early childhood. Indeed, of the three out of 160 PLA2R negative biopsies screened at Mayo, one case was a very young child. Furthermore, one of the two adult cases identified in Paris had a very early onset at the age of one year. In total, 11 Sema3B positive biopsies were identified, of which eight were children or young adults, and in five cases the disease started at of before the age of two years.

In the pediatric cohorts, 6/59 (10%) biopsies stained positive for Sema 3B. After withdrawing the 18 cases of lupus, which is the most frequent cause of MN in this age range, the real prevalence of Sema3B-associated disease among non-lupus patients was 15%, thus making Sema3B the first antigen in pediatric MN. Bright granular staining of Sema3B was seen in all biopsies and was shown to colocalize with IgG staining by confocal immunofluorescence microscopy. Four cases, all pediatric, also showed granular deposits of IgG along the tubular basement membrane which appeared electron dense by electron microscopy. By Western blot with human recombinant Sema3B, circulating anti-Sema3B antibodies were detected in patients with active disease (proteinuria) but unexpectedly, they reacted only with the reduced form of the protein after destruction of the disulfide bonds by reducing agents, which is in sharp contrast with all other MN antibodies that recognize conformational epitopes destroyed by reduction. All other tested sera from controls and patients in remission were negative.

Clinical and biopsy findings showed diverse features of “secondary” MN including association with type-1 diabetes and thrombocytopenic purpura, occasional full-house immunofluorescence, and lack of staining for IgG4 in most cases. Additionally, genetic factors were likely involved in two siblings and in a family where the father and one child were affected with the disease.

Semaphorins are a group of secreted and transmembrane/membrane bound proteins containing a conserved extracellular Semaphorin (sema) domain of about 500 amino acids characterized by highly conserved cysteine residues forming intra-subunit disulfide bonds. Breaking of these disulfide bonds might unmask new protein domains that could feature neoepitopes. The sema domain is the critical component through which semaphorins mediate their effects during development and in adults through interactions with their receptors plexin and neuropilin. Sema3B is a secreted, 81 kDa protein; contrary to Sema3A which was shown to regulate slit diaphragm proteins, the role and function of Sema3B in the kidney is not known [55].

### 4.5. More Antigens on the Horizon: Need for an Antigen-Based Classification and Revisit of the Pathogenesis

As anticipated by Hayashi and Beck [44], two additional antigens, protocadherin-7 (PCDH7) [56] and high temperature recombinant protein A1 (HTRA1) [57], were presented at the last ASN meeting (2020). Others are most probably in the pipeline as 10 to 20% of “primary” MN are still without antigen identified (Figure 3). The question then arises whether identification of these rare forms of MN has a clinical interest and whether the classification of MN between primary and secondary forms that was justified before the identification of the major antigens remains valid. Actually, the whole picture has somewhat changed. For each serological type of MN molecularly defined by an antigen (or the circulating antibodies), some patients have a secondary form while others have a primary form without an etiology as yet specified. Identification of the antigen or biomarker is of the utmost importance because it guides the etiological investigations and because circulating antibodies to those antigens will provide invaluable tools for treatment monitoring once specific assays become available. Consequently, we believe that the time has come to propose a new molecular classification of MN based on the target antigen or biomarker identified. When an etiology is established, treatment should first aim at treating the cause. In the other cases, preliminary data suggest that response to therapy is about the same as in PL2AR-related MN which was expected given the auto-immune nature of the disease.

Most of the “new” antigens are weakly, if at all, expressed in the podocytes. This is notably the case for NELL-1, the second antigen after PLA2R. On the other hand, there is clear evidence that NELL-1 is circulating. Sema3B is also a secreted protein. This raises the question of the pathogenetic role of circulating immune complexes initially formed in the blood and subsequently deposited in the glomerular capillary wall, and of the site of primary immunization. Even for PLA2R, this question has not been solved while there are clues, not the least being the role of pollution, suggesting that the lung might be the trigger organ [58].

## 5. Revisiting the Diagnostic and Therapeutic Algorithms in Light of the Antigenic Revolutions: Toward More Personalized Medicine

### 5.1. Noninvasive Diagnosis of Primary MN

While the diagnosis of “primary” MN required a kidney biopsy until recently, the availability of assays for PLA2R antibodies has induced a dramatic change of the diagnostic strategy, owing to the very high specificity (99%) of anti-PLA2R antibody for the diagnosis of MN [59,60]. The 95% confidence interval for specificity is 0.96 to 1.0, which is comparable to the diagnostic performance of kidney biopsy. The added value of kidney biopsy was studied by the Mayo Clinic investigators in 97 patients who tested positive for PLA2R antibodies and had no evidence of a secondary cause of MN. Sixty of those 97 patients had an estimated glomerular filtration rate (eGFR) >60 mL/min/1.73 m^2^. In these patients, the kidney biopsy did not provide significant information that altered management; one patient had a superimposed diabetic nephropathy, and a second patient had a superimposed focal segmental glomerulosclerosis lesion. Among the 37 patients with primary MN and eGFR <60 mL/min/1.73 m^2^ additional findings included acute interstitial nephritis (*n*=1), acute tubular necrosis (*n* = 1), FSGS (*n* = 2) with cellular crescents in one case, and diabetic nephropathy (*n* = 1). These findings were corroborated by Wiech et al. [61] who showed that only twelve (6%) of the 194 PLA2R1-Ab positive patients with a kidney biopsy, had a second relevant diagnosis in addition to MN: five (3%) patients had interstitial nephritis, five (3%) had a diabetic nephropathy was diagnosed and two (1%) had IgA nephropathy. These patients had a median eGFR of 51 mL/min/1.73 m^2^. Given the potential risks and costs of the biopsy procedure, the current recommendations in PLA2R-Ab positive patients [10] are to consider a kidney biopsy in the patients with unexplained deterioration of eGFR or unusual clinical presentation suggesting a secondary cause (in particular positive antinuclear antibodies), (Figure 4).

In patients without PLA2R-Ab including by immunofluorescence detection, which is a bit more sensitive than the ELISA, a kidney biopsy should be performed to establish the diagnostic of MN and the antigen identified by staining of the paraffin embedded biopsies with specific antibodies against PLA2R, THSD7A, and the recently described antigens, all commercially available (Figure 5). Choice of antibody for biopsy staining should be prioritized according to the clinical context: age, systemic manifestations, extent of the Ig deposits (segmental or global).

Staining for antigen can be performed by immunohistochemistry (IHC) or immunofluorescence (IF) on deparaffinized sections of the kidney biopsy.

For each antigen, the main etiology or medical setting is indicated.

### 5.2. More Personalized Treatment Based on Antibody Monitoring

The development of ELISA for quantitative assessment of PLA2R-Ab has induced a paradigm shift in the monitoring of patients with PLA2R MN. Before the discovery of PLA2R-Ab [35], daily proteinuria, changes in eGFR, and other complications of the disease were the criteria to assess the efficacy of treatment in the absence of a reliable biomarker. Although daily proteinuria is still the major parameter that defines remission, it is now well established that in most patients with PLA2R related MN, the decrease of PLA2R-Ab precedes remission by weeks or months [62,63,64]; conversely, their re-appearanceor their substantial increase predicts relapse. The lag time of several months, sometimes more than a year, from immunological remission to clinical remission is caused by the glomerular capillary lesions that need to be repaired.To evaluate treatment response and adjust therapy, the KDIGO 2021 Guideline for the Management of Glomerular Diseases [10] will recommend longitudinal monitoring of PLA2R-Ab levels with a first assessment at six months after start of therapy. They define complete immunological remission by a cut-off value of 2 RU/mL by ELISA; a negative IF test also indicates immunological remission. Some centers measure PLA2R-Ab at three months and adapt treatment at that time, based on the results of observational studies and the GEMRITUX trial [22] showing that the maximum efficacy of treatment was reached at 3 months with no further decrease of PLA2R-Ab between three and six months. Results obtained in THSD7A related MN also suggest that monitoring of antibodies has a predictive value [65]. If complete immunological remission is achieved, rituximab or cyclophosphamide should be interrupted and CNI progressively tapered.

For the most recent antigens, preliminary results obtained by Western blot suggest that circulating levels of the relevant antibodies are correlated with disease activity [41,42]. ELISAs are not available yet. Given the rarity of the new antigens (except for NELL-1), the development of chip assays both for the detection and the monitoring of antibodies is most likely. They will be mandatory for personalized treatment monitoring.

## 6. Conclusions

The years 2019–2020 have seen substantial advances in the knowledge of MN which translate in the daily care. Although MENTOR, STARMEN and RI-CYCLO are important steps toward better patients’ care, future studies are needed to define the best treatments with close monitoring of PLA2R-Ab to finely tune treatment choice and posology. Beyond PLA2R, the discovery of new antigens already has a profound impact on diagnostic algorithm, search for etiology, and very soon treatment monitoring when assays to quantify the relevant antibodies will be available.

## Figures and Tables

**Figure 1 jcm-10-00607-f001:**
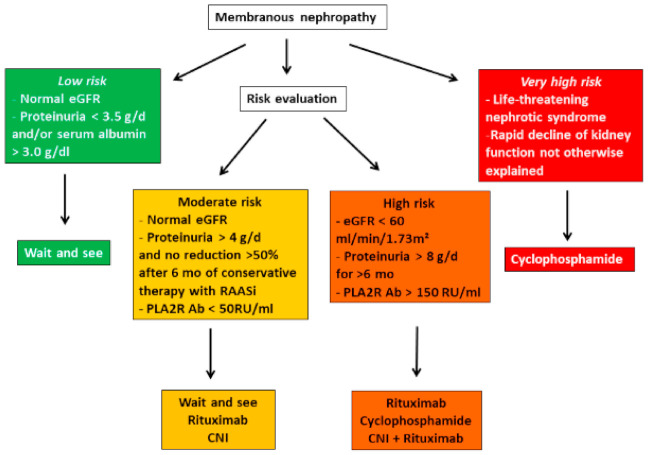
Risk-based treatment of membranous nephropathy (adapted from Clinical Practice Guidelines for the management of glomerulonephritis [10]).

**Figure 2 jcm-10-00607-f002:**
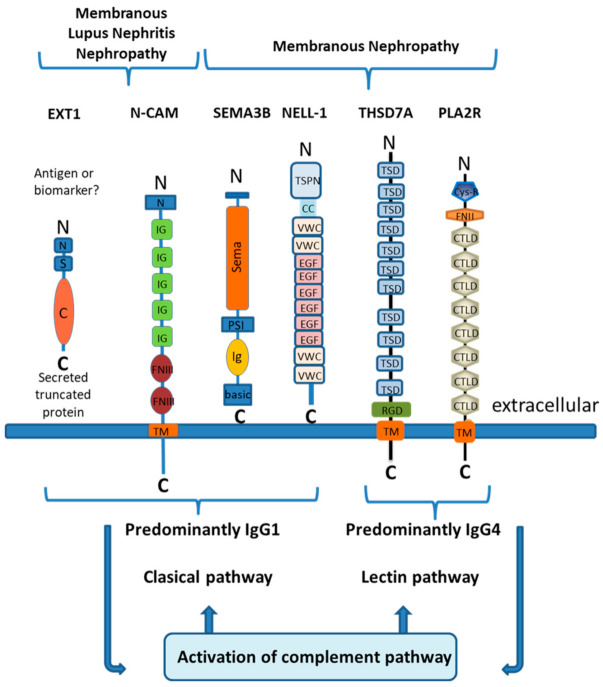
Antigens of membranous nephropathy and their schematic molecular structure.Exostosins are transmembrane proteins in the Golgi apparatus that have a short amino-terminal cytoplasmic tail (N), a single transmembrane domain (S), and a long globular catalytic C-terminal domain(C) within the Golgi lumen.NCAM1contains an amino-terminal tail (N), five immunoglobulin domains (IgG), two fibronectin type III domains (FNIII), a transmembrane domain (TM), and an intracellular region at the C terminal end.Sema3Bhas large Sema domain region, a plexin-semaphorin-integrin (PSI) domain, an Ig domain, and a short C terminal basic domain.NELL1 is characterized by an amino-terminal TSP-1-like (TSPN) domain, a coiled-coil (C-C) domain, two von Willebrand factor type C (VWC) domains, six EGF-like domains (E), and two VWC domains at the cytoplasmic end.THSD7A contains 11 trombospondin type-1 domains (TSD),an arginine-glycine-aspartic acid motif (RDG),a transmembrane domain(TM) and a C terminalintracellular end.PLA2R contains a cys-rich domain (Cys-R), a fibronectin type II domain (FNII),8 C-type lectin-like domains (CTLD),a transmembrane domain (TM), and a C terminal intracellular end. Shown is the predominant Ig subclass of the relevant antibodies and the main activated complement pathway.

**Figure 3 jcm-10-00607-f003:**
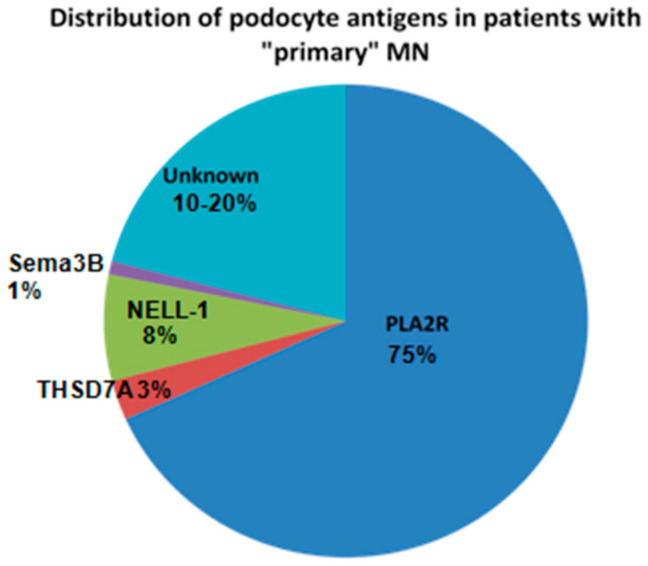
Distribution of podocyte antigens in patients with “primary” membranous nephropathy (MN).

**Figure 4 jcm-10-00607-f004:**
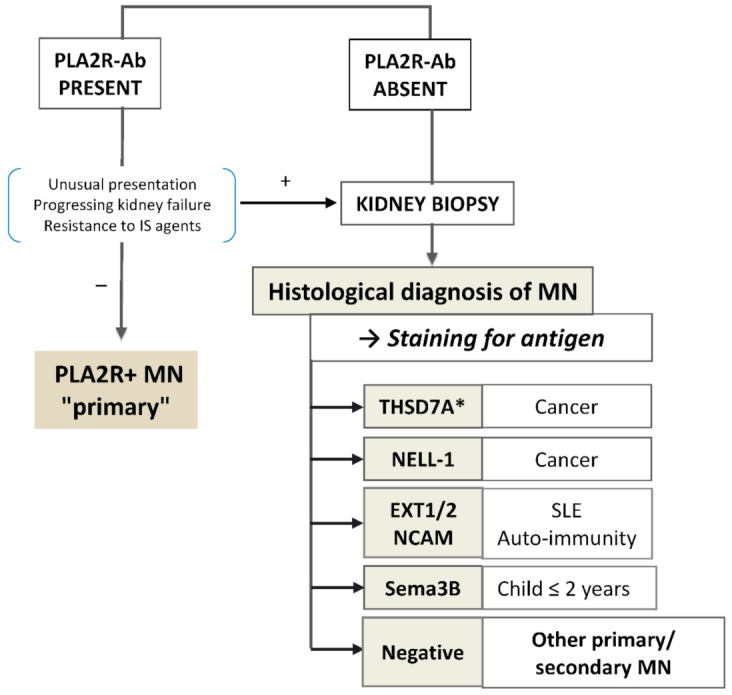
Simplified diagnostic algorithm. In patients with PLA2R-Ab but with unusual presentation or deteriorating renal function or resistant to IS agents, a kidney biopsy is indicated (+); it is not mandatory in the other cases (−). *An immunofluorescence test for the detection of circulating anti-THSD7A antibodies is available but as yet there is no consensus on whether a kidney biopsy should be performed or not in case of positive serology.

**Figure 5 jcm-10-00607-f005:**
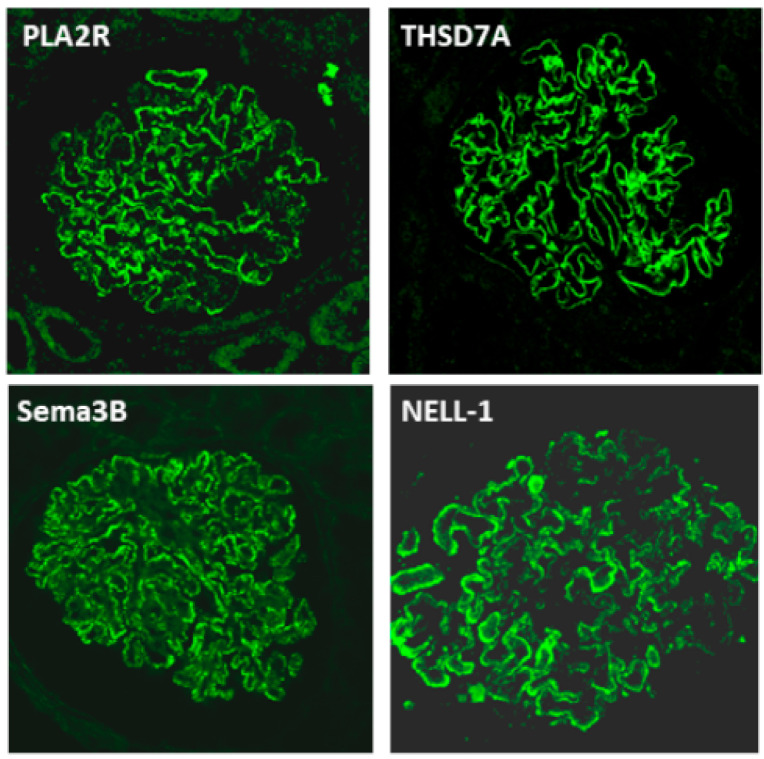
Immunofluorescence staining of the four antigens: PLA2R, THSD7A, Sema3B, NELL-1. Paraffin embedded tissue biopsies were stained after retrieval with the relevant antibodies. Fine granular staining of the immune deposits is seen on the outer aspect of the glomerular basement membrane. Note that some capillary loops are not, or weakly stained with anti-NELL-1 antibody.

**Table 1 jcm-10-00607-t001:** Major randomized clinical trials (RCTs) on membranous nephropathy since 2019.

Study	MENTOR [7]	STARMEN [8]	RI-CYCLO [9]
Design	Multicenter North America *N* = 130	Multicenter Europe *N* = 86	Multicenter Italy+Switzerland *N* = 74
Inclusion	Proteinuria > 5 g/24 h CrCI > 40 mL/min/1.73 m^2^	Proteinuria > 4 g/24 h eGFR > 45 mL/min/1.73 m^2^	Proteinuria > 3.5 g/24 h eGFR ≥ 30 mL/min/1.73 m^2^
PLA2R positivity	74% (96/130)	77% (53/69)	66% (59/73)
Run-in	3 months	6 months	3 months
Comparison	Rituximab vs. Cyclosporine	Modified Ponticelli (Methylprednisolone + cyclophosphamide) vs. tacrolimus + rituximab	Modified Ponticelli (Methylprednisolone + cyclophosphamide) vs. Rituximab
Remission definition	CR: proteinuria < 0.3 g/24 h, Alb > 3.5 g/dL PR: proteinuria 50% reduction from baseline + range between 0.3–3.5 g/24 h Relapse: proteinuria > 3.5 g/24 h after CR or PR	CR: proteinuria < 0.3 g/24 h, eGFR > 45 mL/min PR: same as MENTOR, eGFR > 45 mL/min Relapse: same as MENTOR	Same as MENTOR
Outcome CR + PR (CR only)	12 months		
	60% vs. 52%	79% vs. 51%	73% vs. 62% (32% vs. 16%)
	24 months		
	60% vs. 20% (35% vs. 0%)	84% vs. 58% (60% vs. 26%)	81% vs. 85% (35% vs. 42%)
SAE	17% vs. 31%	19% vs. 14%	14% vs. 19%

Results of outcome are given by Intention-To-Treat; PR, partial remission; SAE, severe adverse events.

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
