# Peer review of "Advances in Membranous Nephropathy"

_jcm, 2021, doi:10.3390/jcm10040607_

Round 1

Reviewer 1 Report

This review by Ronco et al. is a well-written summary of the latest discoveries in the treatment and pathophysiology pertaining to MN. This manuscript provides a clear and well-organized overview of new advances in diagnosis and therapy of MN. 

  1. I would reformulate the sentence beginning by “Thanks” in the abstract on page 1, line 20. “Given the technological leap…” would be an appropriate alternative.
  2. Since several MN-related antigens have been discovered in recent years, I find the statement that 80% of cases occurred in “the absence of cause” a bit too strong (page 1, line 38).
  3. The long-term renal toxicity of CNIs might also be added as a potential cause for favoring anti-CD20 therapy (page 2, line 62). Pre-existing kidney scarring is often the main reason not initiating CNIs.
  4. It would be useful to state the % of anti-PLA2R patients enrolled in MENTOR on page 2 to get a sense of the population studied.
  5. When the authors comment results from the STARMEN trial (page 3, starting on line 121), they do not mention that the superiority of corticosteroid-cyclophosphamide over tacrolimus-rituximab might be related to the relatively small dose of rituximab given to patients (only 1 g, instead of 2 g or even 4 g in total in MENTOR). My experience of using PLA2R-based treatment algorithm tends to demonstrate that rituximab should be often given several times to reach immunological remission.
  6. Figure 1 refers to “Table MN1” for risk evaluation, but this table is not integrated to the manuscript. I suggest that the authors add definitions of low risk, moderate risk and high risk patients to the figure for more clarity.

Author Response

Reply to Reviewer-1

This review by Ronco et al. is a well-written summary of the latest discoveries in the treatment and pathophysiology pertaining to MN. This manuscript provides a clear and well-organized overview of new advances in diagnosis and therapy of MN. 

We thank the Reviewer for his/her positive evaluation. This is well appreciated.

  1. I would reformulate the sentence beginning by “Thanks” in the abstract on page 1, line 20. “Given the technological leap…” would be an appropriate alternative.

We have reformulated the sentence as requested.

  1. Since several MN-related antigens have been discovered in recent years, I find the statement that 80% of cases occurred in “the absence of cause” a bit too strong (page 1, line 38).

We have replaced "absence of cause " by "absence of associated disease". The absence of associated, potentially causative disease defines primary MN.

  1. The long-term renal toxicity of CNIs might also be added as a potential cause for favoring anti-CD20 therapy (page 2, line 62). Pre-existing kidney scarring is often the main reason not initiating CNIs.

We have mentioned this limitation of the use of CNIs as suggested.

It would be useful to state the % of anti-PLA2R patients enrolled in MENTOR on page 2 to get a sense of the population studied.

We have now mentioned that   74 % of the patients enrolled in MENTOR had PLA2R-Ab.

When the authors comment results from the STARMEN trial (page 3, starting on line 121), they do not mention that the superiority of corticosteroid-cyclophosphamide over tacrolimus-rituximab might be related to the relatively small dose of rituximab given to patients (only 1 g, instead of 2 g or even 4 g in total in MENTOR). My experience of using PLA2R-based treatment algorithm tends to demonstrate that rituximab should be often given several times to reach immunological remission.

Again we fully agree with the Reviewer's comments. This limitation is now mentioned in the descrition of the results of MENTOR.

Figure 1 refers to “Table MN1” for risk evaluation, but this table is not integrated to the manuscript. I suggest that the authors add definitions of low risk, moderate risk and high risk patients to the figure for more clarity.

Thank you very much for the suggestion. We have completed Figure 1 with the definitions of the risk class.

Reviewer 2 Report

General Comment: This is an excellent review of recent advances, of which there have been many in the past few years. The authors are experts in the field and provide excellent perspectives on the topic. The review is timely and most needed, filling a critical void in the current literature. Suggestion are detailed below and are generally minor. On balance, the pathophysiology section is a lot more in depth compared to review of therapeutic advances and the more personalized treatment based on antibody monitoring at the end. The Abstract suggest more of a focus on therapeutics. Suggest making the pathophysiology section more succinct or summarize more of the key points in Abstract. The authors may consider expanding on the “more personalized treatment based on antibody monitoring”. Small amounts of edits are needed for grammar and to remove redundancy to make more concise.

Introductory paragraphs (pages 1-2): Based on the introductory paragraphs, the focus of the review is recent advances of the last two years. It would thus be more informative to provide a global summary of the understanding of MN up to the last two years to provide a foundation of understanding prior in-depth review of the trials and studies published in the last two years. More specifically, characterization such as “antigens that have long eluded identification (line 34)” and “absence of an established cause (line 38)” is somewhat outdated. It does not set up the readers to understand the focus on anti-CD20 therapies in the recent trials and the focus on anti-PLA2R antibody titers, so the flow into the next sections does not work very well.

KDIGO recommendations (page 6): consider providing definition for low risk and moderate risk groups.

Figure 1. In this PDF version, figure is scrambled. Title, suggest indicating that this is primary membranous nephropathy. Recommend including more details either directly in the table or in the footnotes so that the information is complete and figure can be well-understood on its own. Making reference to “see Table MN1” makes the figure a lot less useful to readers.

Improving the use of anti-CD20 antibodies (Page 8): “lower risk of immunization” – unclear what this means.

New therapeutic approaches: anti-Baff therapy (belimumab), anti-plasma cell therapy, 92 immunoadsorption, anti-complement therapy (Page 9): One sentence was devoted to “immunoadsorption and plasmapheresis”. Can the authors include a more in-depth discussion on comparative efficacy of these therapies or issues / uncertainties surrounding these therapies so that this is more informative to the readers? Last paragraph in this section (Line 120-127), include citation or references to the early phase trials?

Exostosins 1&2 (EXT1/2) and NCAM-1 are associated with auto-immune diseases (Page 10): Line 191-192 Among other hypotheses, reactivity… only present in the glomerular “enzyme”. What is “glomerular “enzyme”?

Neural epidermal growth factor-like 1 protein (NELL-1) is associated with "primary" 228 MN and cancer-related MN (Page 11-12): Lines 264 – 270, the numbers of biopsies positive for NELL1/THSD71/PLA2R versus the prevalence by percent do not work out mathematically. Lines 279-285: or overexpression of NELL-1 by malignant cells and subsequent deposit in the kidney, citation for the suggestion of “external stimuli or epigenetic modification”?

Semaphorin 3B (Sema3B) is associated with early childhood MN (page 12): First sentence is not clear. Lines 301-305 – unclear the meaning and significance of reaction with the “reduced form” – please explain.

Figure 2: Abbreviations LMN V and MN needs to be described in footnotes. Consider including key to explain what structures the shapes/colors represent.

Figure 3: Recommended a more informative figure title.

More personalized treatment based on antibody monitoring (page 16-17): Line 400, “daily proteinuria was the only criterion to assess the efficacy of treatment”: statement much too simplistic. Of course the assessment of response is based on proteinuria, changes in eGFR, and other complications of the disease. Perhaps highlight the key difference that antibodies offer a way to predict disease activity and prevent recurrence.

Line 408-409: “based on thre results of … no further benefit between 3 and 6 months” – sentence is unclear. Are the authors attempting to discuss following antibody levels at 3 versus 6 months? What is the recommendation?

This section is more general and not as in-depth as other section. 

Author Response

Reply to Reviewer-2

Comments and Suggestions for Authors

General Comment: This is an excellent review of recent advances, of which there have been many in the past few years. The authors are experts in the field and provide excellent perspectives on the topic. The review is timely and most needed, filling a critical void in the current literature. Suggestion are detailed below and are generally minor. On balance, the pathophysiology section is a lot more in depth compared to review of therapeutic advances and the more personalized treatment based on antibody monitoring at the end. The Abstract suggest more of a focus on therapeutics. Suggest making the pathophysiology section more succinct or summarize more of the key points in Abstract. The authors may consider expanding on the “more personalized treatment based on antibody monitoring”. Small amounts of edits are needed for grammar and to remove redundancy to make more concise.

Thank you very much for your positive comments much appreciated. We have taken the option to summarize more of the key points in the abstract and we have expanded the chapter on personalized treatment at the end.

Introductory paragraphs (pages 1-2): Based on the introductory paragraphs, the focus of the review is recent advances of the last two years. It would thus be more informative to provide a global summary of the understanding of MN up to the last two years to provide a foundation of understanding prior in-depth review of the trials and studies published in the last two years. More specifically, characterization such as “antigens that have long eluded identification (line 34)” and “absence of an established cause (line 38)” is somewhat outdated. It does not set up the readers to understand the focus on anti-CD20 therapies in the recent trials and the focus on anti-PLA2R antibody titers, so the flow into the next sections does not work very well.

Thank you very much for your great suggestion. We have changed part of the introduction accordingly.

KDIGO recommendations (page 6): consider providing definition for low risk and moderate risk groups.

Thank you very much for the suggestion. We have completed Figure 1 with the definitions of the risk class.

Figure 1. In this PDF version, figure is scrambled. Title, suggest indicating that this is primary membranous nephropathy. Recommend including more details either directly in the table or in the footnotes so that the information is complete and figure can be well-understood on its own. Making reference to “see Table MN1” makes the figure a lot less useful to readers.

The figure was not scrambled in the submitted MS. We will send a note to  the publisher to avoid this. We have completed Figure 1 with the definitions of the risk class.

Improving the use of anti-CD20 antibodies (Page 8): “lower risk of immunization” – unclear what this means.

The 3 "new" antibodies are humanized, hence the lower risk of immunization against the monoclonal antibody

New therapeutic approaches: anti-Baff therapy (belimumab), anti-plasma cell therapy, 92 immunoadsorption, anti-complement therapy (Page 9): One sentence was devoted to “immunoadsorption and plasmapheresis”. Can the authors include a more in-depth discussion on comparative efficacy of these therapies or issues / uncertainties surrounding these therapies so that this is more informative to the readers? Last paragraph in this section (Line 120-127), include citation or references to the early phase trials?

Only very small series or cases of patients treated with immunoadsorption or plasmapheresis have been published. For the time being, we think the only indication is refractory disease and this should be discussed with a reference center.

The last paragraph is devoted to anti-complement therapy for which there is no published report to the best of my knowledge.

Exostosins 1&2 (EXT1/2) and NCAM-1 are associated with auto-immune diseases (Page 10): Line 191-192 Among other hypotheses, reactivity… only present in the glomerular “enzyme”. What is “glomerular “enzyme”?

By "glomerular" enzyme, we mean the enzyme produced in the glomerulus possibly by podocytes while we used a recombinant enzyme produced in human eukaryotic cells  in our assays.

Neural epidermal growth factor-like 1 protein (NELL-1) is associated with "primary" 228 MN and cancer-related MN (Page 11-12): Lines 264 – 270, the numbers of biopsies positive for NELL1/THSD71/PLA2R versus the prevalence by percent do not work out mathematically. Lines 279-285: or overexpression of NELL-1 by malignant cells and subsequent deposit in the kidney, citation for the suggestion of “external stimuli or epigenetic modification”?

We clarified the first point by adding " When the prevalence of cancer was analyzed per subcategory of MN identified by antigen, NELL1 came first....."

We have added a reference for epigentic modifications of PLA2R1 in cancer cells.(Semin Cancer Biol doi: 10.1016/j.semcancer.2017.11.002. Epub 2017 Nov 2.Current insights into functions of phospholipase A2 receptor in normal and cancer cells: More questions than answers Olga Sukocheva et al)

Semaphorin 3B (Sema3B) is associated with early childhood MN (page 12): First sentence is not clear. Lines 301-305 – unclear the meaning and significance of reaction with the “reduced form” – please explain.

We specified "in the cohorts screened at Mayo Clinics and in Paris mostly composed of adult patients"

We specified "reduced form of the protein after destruction of the disulfide bonds by reducing agents"

Figure 2: Abbreviations LMN V and MN needs to be described in footnotes. Consider including key to explain what structures the shapes/colors represent.

We have followed your recommendations. Abbreviations are now described as well as the antigen structures.

Figure 3: Recommended a more informative figure title.

The title was Distribution of podocyte antigens in patients with "primary" MN. We apologize that the the title was above the piechart and not after "Figure 3".

More personalized treatment based on antibody monitoring (page 16-17): Line 400, “daily proteinuria was the only criterion to assess the efficacy of treatment”: statement much too simplistic. Of course the assessment of response is based on proteinuria, changes in eGFR, and other complications of the disease.

The Reviewer is right. we have added  changes in eGFR, and other complications of the disease.

Perhaps highlight the key difference that antibodies offer a way to predict disease activity and prevent recurrence.

The reviewer is correct. Antibodies are tightly correlated to disease activity , their decrease announces remission while their re-appearance or re-increase predicts relapse. Consequently, we have emphasized the importance of antibodies for disease monitoring.

Line 408-409: “based on thre results of … no further benefit between 3 and 6 months” – sentence is unclear. Are the authors attempting to discuss following antibody levels at 3 versus 6 months? What is the recommendation?

The Reviewer is correct. In most patients, response occurs within 3 months after the start of therapy. This is the reason why some centers will measure anti-PLA2R antibodies at 3 months, and adapt treatment at that time. The KDIGO guidelines are more conservative, recommending a first evaluation of anti-PLA2R antibodies at 6 months, but also recognize that some centers will assess antibodies earlier.

This section is more general and not as in-depth as other section. 

We have developed this part for PLA2R-associted MN as requested.